# Metabolomics-Based Analyses of Dynamic Changes in Flavonoid Profiles in the Black Mulberry Winemaking Process

**DOI:** 10.3390/foods12112221

**Published:** 2023-05-31

**Authors:** Yanan Qin, Haotian Xu, Ya Chen, Jing Lei, Jingshuai Sun, Yan Zhao, Weijia Lian, Minwei Zhang

**Affiliations:** 1Xinjiang Key Laboratory of Biological Resources and Genetic Engineering, College of Life Science & Technology, Xinjiang University, Urumqi 830046, China; qingyalan12345@sina.com (Y.Q.); xuhaotian0202@163.com (H.X.); sjs201534@163.com (J.S.); zhaoyan980818@sina.com (Y.Z.); 2Turpan Institute of Agricultural Sciences, Xinjiang Academy of Agricultural Sciences, Turpan 838000, China; chenya1107@sina.com (Y.C.); leijing200288@163.com (J.L.); 1634321773lwj@sina.com (W.L.)

**Keywords:** mulberry wine, fermentation, metabolomics, UHPLC-QE-MS/MS, flavonoids

## Abstract

To overcome the fruit’s perishability, mulberry wine has been developed as a method of preservation. However, dynamic changes in metabolites during mulberry wine fermentation have not been reported yet. In the present investigation, UHPLC-QE-MS/MS coupled with multivariate statistical analyses was employed to scrutinize the metabolic profiles, particularly the flavonoid profiles, throughout the process of vinification. In general, the major differential metabolites encompassed organic heterocyclic compounds, amino acids, phenylpropanoids, aromatic compounds, and carbohydrates. The contents of total sugar and alcohol play a primary role that drove the composition of amino acids, polyphenol, aromatic compound, and organic acid metabolites based on the Mantel test. Importantly, among the flavonoids, abundant in mulberry fruit, luteolin, luteolin-7-O-glucoside, (−)-epiafzelechin, eriodictyol, kaempferol, and quercetin were identified as the differential metabolic markers during blackberry wine fermentation and ripening. Flavonoid, flavone and flavonol biosynthesis were also identified to be the major metabolic pathways of flavonoids in 96 metabolic pathways. These results will provide new information on the dynamic changes in flavonoid profiles during black mulberry winemaking.

## 1. Introduction

Mulberry fruit (*Morus nigra* L.) is widely cultivated in diverse climatic and geographic conditions due to its excellent sensory and nutritional characteristics [1]. It contains a rich and diverse range of bioactive compounds. Polyphenols, particularly flavonoids, are functionally significant compounds that are abundant in both type and quantity in mulberry [2]. However, the flavonoid profile of mulberry undergoes dynamic changes due to various technological factors during processing into value-added products such as juice, jam, and wine [3]. Hence, given the high-value nature of mulberry wine and its potential health benefits [4], studying the evolution of flavonoid compounds during its fermentation process is crucial.

Fermentation and ripening are two key technologies in involved in the transformation of flavonoid compounds during mulberry wine production. During fermentation, microbial communities play a crucial role in mediating biotransformation through cell wall absorption, hydrolysis of flavonoid glycosides, and the formation of new metabolites [5]. Lactic acid bacteria and yeast utilize glucosidases, cellulases or hemicelluloses, and esterase to soften the plant tissue structure and release esterified nutrients [6,7]. Along with changes in environmental factors, such as pH, total sugar, total acid, and assimilable nitrogen, microorganisms enhance the utilization of fermentation substrates during their growth period through glycosylation, DE-glycosylation, ring cleavage, methylation, glucuronidation, and sulfation pathways [8]. While changing the distribution of flavonoid compounds, the flavonoid glycosides are further reduced to flavonoid aglycones under the influence of environmental factors such as pH [9]. It is the combined action of the fermentation environment and microorganisms that promote the transformation of flavonoid compounds and leads to their reduced content [6,10].

The number of hydroxyl groups on the C_6_-C_3_-C_6_ structure determines the hydrolysis properties of flavonoids [11]. During ripening, the hydrolysis of the b-ring and c-ring of flavonol glycosides leads to the reduction of flavonol glycosides [12]. Moreover, hydroxyl groups on the C skeleton can be replaced by lipophilic compounds or esterified with fatty acids [13,14], which changes the polarity and solubility of flavonoid molecules and forms insoluble polymers such as polymethoxyflavones [15]. With prolonged ripening time, in a relatively stable environment, a variety of flavonol glycosides from other flavonoid aglycones and/or triglycosides may accumulate, losing one or more sugar residues through glycosidase-mediated hydrolysis [16]. These reasons further lead to a decrease in flavonoid content. Compared with the fermentation process, the ripening process involves fewer complex biochemical reactions and more incubation of flavonoid stability.

In the present study, utilizing ultrahigh-performance liquid chromatography (UHPLC) coupled with Q Exactive mass spectrometry (MS), we assessed the metabolic profile alterations that take place during the process of fermentation. The correlation between polyphenolic compounds and major environmental factors was accurately shown through our analytical methodology. Variation in and metabolic pathways of flavonoids were comprehensively described, which serve as biomarkers in mulberry wine. Our findings provide a robust theoretical foundation for the optimization of the fermentation process and the preservation of the quality of mulberry wine, particularly with regard to the evolution of flavonoids.

## 2. Materials and Methods

### 2.1. Chemicals and Reagents

All chemicals were of analytical grade unless stated otherwise. Methanol and acetonitrile were purchase from CNW Technologies, Germany, ammonium acetate (SIGMA-ALDRICH, St. Louis, MO, USA), ammonium hydroxide (Fisher Chemical, Riverside, CA, USA), ddH_2_O (Watsons, China). HCl (1 + 1 (*v*/*v*)), NaOH (200 g/L), glucose standard solution (2.5 g/L), methylene blue (10 g/L), phenolphthalein, CuSO_4_ (0.05 g/mL), and NaOH (0.05 mol/L) were bought in Macklin (Shanghai, China). The dry yeast (*Saccharomyces cerevisiae* DV 10) was purchased from JATOU (Shanghai, China).

### 2.2. Collection of Samples and Metabolite Extraction

The samples were collected during the winemaking process. The alcoholic fermentation usually lasts for 12 days and the ripening in the storage tank takes about six months. Therefore, the samples were collected from the tanks at 12 days and six months. The crushed black mulberry musts and black mulberry wines were collected during the corresponding process. For each process, 9 samples (3 samples of each layer; the tanks were divided into a high layer, middle layer, and bottom layer) were mixed thoroughly. Later, each mixed sample was divided into three parts and stored at −80 °C for physicochemical properties and UHPLC-QE-MS/MS analysis. For the sake of convenience and easy reference in this study, the abbreviations J, F, A, and P are used for the crushed black mulberry musts, fermented musts, ripening musts, and black mulberry wines, respectively.

Each sample (100 μL) was mixed with 400 μL of extract (acetonitrile:methanol = 1:1, containing an isotopically labeled internal standard mixture) in an EP tube and subsequently stood at −20 °C for 2 h. Next, the samples were vortexed for 30 s, sonicated for 10 min in an ice-water bath, and incubated for 1 h at −40 °C to precipitate proteins, then centrifuged at 12,000 rpm for 15 min at 4 °C, after which the supernatant was transferred to a new glass vial for further analysis.

### 2.3. Mulberry Winemaking

Black mulberry fruit (*Morus nigra* L.) was harvested through handpicking from the Turpan area Fruiter Experiment Centre, Xinjiang Province, China, on 15 May 2022. Briefly, the fruit was crushed by belt press (LDD1500P_2–4_, 4000 × 2300 × 2400, China), and then sulfur dioxide (0.03 g/kg) and pectin (0.225 g/kg) were added to the musts with enzymolysis at 32 °C for 3 h and poured into the fermentation tank. Subsequently, 0.25 g/L of activated *Saccharomyces cerevisiae* DV10 was purchased from LALVIN, Denmark and added to the musts at the bottom of the fermentation tank. Alcoholic fermentation was carried out at 18 °C in a 1000 L fermentation tank. According to the conventional red wine production process, fermentation monitoring, chaptalization, decanting, and yeast nutrition (MERMAID E) addition were performed within the whole fermented process until the alcohol content of the must reached 8 *v*/*v*, %. Sulfur dioxide (0.05 g/kg) was used to stop the fermentation. Thereafter, the fermented musts were kept at 5 °C for 10 days for natural clarification, so the pulp and other suspended substances could settle down to the bottom of the tank. The supernatants were extracted from the storage tank and stored at 5 °C for at least 6 months of ripening. Finally, pasteurization was performed at 68 °C (30 min), and black mulberry wines were bottled in volumetric glass jars. The whole process is depicted in Figure 1.

### 2.4. Analysis of Physicochemical Property

The physicochemical properties of the J, F, A and P samples were determined, including the pH, total acid, reducing sugar, total soluble solids, and alcohol contents, respectively. pH was measured with a digital benchtop pH meter (PHS-3C, Shanghai, China). The determination of total acid (TA) content was achieved through potentiometric titration (g/L tartaric acid), using a pH of 8.2 as the endpoint. The quantification of total sugar (TS) content was accomplished via direct titration. Additionally, the measurement of total soluble solids (Brix) was conducted utilizing a handheld Brix meter, while the determination of alcohol content was executed using the alcohol meter method (M277465, China). All treatments were executed in triplicate within the laboratory, and the outcomes are expressed as means ± standard deviation.

### 2.5. UHPLC-QE-MS/MS Analyses and Data Processing

A Vanquish UHPLC system (Thermo Fisher Scientific, Waltham, MA, USA) with a UPLC BEH Amide column (2.1 mm × 100 mm, 1.7 μm), coupled with an Q Exactive HFX mass spectrometer (Orbitrap MS, Thermo Fisher Scientific, Waltham, MA, USA) was employed to analyze the metabolic profiling. The mobile phase consisted of 25 mmol/L of ammonium acetate and 25 mmol/L of ammonia hydroxide in water (pH = 9.75) (A) and acetonitrile (B), respectively. The autosampler temperature was 4 °C and the injection volume was 2 μL. All samples had 5 replicates to ensure the reproducibility and stability of the instrument analysis. Quality control (QC) samples were also added to monitor the stability of the system to ensure reliable experimental data. The conditions of the QE HFX mass spectrometer were as follows. Ion source: Electrospray ionization (ESI) (Thermo Fisher Scientific, Waltham, MA, USA); ionization mode: positive and negative mode; mass spectrometry parameters: Intrathecal gas flow rate 30 Arb; auxiliary gas flow rate 25 Arb; capillary temperature 350 °C; full MS resolution 60,000; mass range: 70–1050; analysis time: 12 min; gradient of separation: 0 min, 0% B; 0–8 min linear increase from 0 to 100% B; 8–10 min hold at 100% B; 10–12 min return to 0 B and re-equilibrate the column. MS/MS resolution 7500; collision energies of 10, 30, 60 with normalized NCE mode and injection voltage of 3.6 kV (positive) or −3.2 kV (negative).

The ProteoWizard program and an in-house program developed using R were employed to generate the LC-MS/MS data. The raw data for the positive ion mode consisted of 3 quality control (QC) samples and 20 experimental samples, from which 7081 peaks were extracted. In order to reduce the impact of the error of the testing system on the results to reduce the impact of systematic errors on the results and to better highlight the biological significance of the results, a series of data management was conducted. The analyses included peak detection, extraction, alignment, integration, and normalization. The cutoff value for annotation was set to 0.3 and raw data (positive and negative ion models) containing chemical substances, *m*/*z*, and peak retention times (RT), as well as peak intensities were obtained. Internal standards were applied to check the quality of the data QC.

### 2.6. Enrichment Analysis Based on Metabolic Pathways

The Kyoto Encyclopedia of Genes and Genomes (KEGG) pathway database (http://www.KEGG.jp/KEGG/pathway.HTML, accessed on 20 March 2023) was employed to explore the metabolic pathways throughout the black mulberry winemaking process. The apparently different metabolites found in the black mulberry wine samples were mapped to the KEGG pathways. Later, the differential metabolites were used to perform enrichment analysis and find the metabolic pathways.

### 2.7. Statistical Analysis

Multivariate analysis was performed using the SIMCA 16.0.2 package (Sartorius Stedim Data Analytics AB, Umea, Sweden). The data were scaled and logarithmically transformed to minimize the noise and high variance effect variables. After these transformations, principal component analysis (PCA) was performed to reduce the data dimension by visualizing the distribution and grouping of samples. Later, the supervised orthogonal projection-discrimination and application analysis (OPLS-DA) on potential structures was performed to visualize the group separation and discover the significantly changed metabolites. Additionally, in order to evaluate the importance of variables in the projection, variable importance in projection (VIP) was determined for the initial principal component in the orthogonal partial least squares discriminant analysis (OPLS-DA). This technique enabled the contribution of each variable to be summarized and identified. The metabolites exhibited VIP > 1, and with a statistical significance level (Student’s *t*-test) of less than 0.05, were considered to be significantly altered metabolites. Later, the Mantel test was performed to explore the relationship between the physicochemical properties and the major metabolite cluster (9999 permutations) using the R package. Finally, the commercial databases KEGG (http://www.genome.jp/kegg/; accessed on 8 May 2023) and MetaboAnalyst (http://www.MetaboAnalyst.ca/; accessed on 8 May 2023) were used for path enrichment analysis.

## 3. Results and Discussion

### 3.1. Physicochemical Parameter Analysis

The changes in pH, total acid (TA), total sugar (TS), total soluble solids (TSS), and alcohol concentration during the J, F, A, and P phases of mulberry wine are presented in Table 1. The pH (5.48), TA (2.793 g/L), TS (139.152 g/L), and TSS (16.9%) of crushed mulberry juice at the initial stage provided adequate conditions to promote the growth and reproduction of microorganisms during the subsequent fermentation processes [17,18]. The pH had decreased to 3.917 and TS also dropped to 39.974 g/L at the end of fermentation. This may be attributed to the consumption of carbohydrates and other macromolecules by lactic acid bacteria during the fermentation process, resulting in the production of organic acids such as lactic and acetic acid [19]. TA had increased to 12.757 (g/L tartaric acid) at the end of fermentation due to the production of a large number of organic acids, but it decreased after the sterilization with pasteurization and the addition of a large amount of SO_2_ during the ripening process, maintaining relative stability thereafter. TSS decreased by 4.47% from smashing to the bottling of mulberry wine due to the influence of microorganisms and environmental conditions, indicating a high utilization of soluble macromolecular organic matter during the fermentation process [20]. The alcohol content gradually increased to 8 *v*/*v*, %during alcohol fermentation and remained stable thereafter (10 *v*/*v*, %). Previous studies have shown that variations in physicochemical factors can facilitate significant organic compound transformations during the fermentation process [21]. Therefore, the utilization of multivariate statistical analysis to explore the evolution of metabolites is of paramount importance.

### 3.2. Multivariate Statistical Analysis

The potential metabolites were identified in different fermentation stages of mulberry wine, including the J, F, A, and P stages. A total of 9699 metabolite ion signatures, including 5879 and 3820 peaks in the positive (ESI+) and negative (ESI−) ion modes, were generated, respectively. Multivariate statistical analyses, including PCA and OPLS-DA models, were performed for the metabolomics data set to reduce dimensionality and visualize sample grouping and enhance data interpretability and effectiveness. The PCA analysis showed that the metabolites involved in the J, F, A, and P stages could be separated and differentiated in the ESI+ and ESI− modes (Figure 2). All samples and QC were within a 95% confidence interval. 78.4% and 72.8% of the differences between ESI+ and ESI− model samples were explained by these results. Especially, the PCA analysis between J versus F, F versus A, A versus P, and J versus P revealed that 87.6%, 88.8%, 77.9%, 91.8%, and 88.5%, 85.2%, 62.4%, 92.1% of the differences were explained by ESI+ and ESI− patterns, respectively (Appendix A). There were significant differences between the unfermented black mulberry juice (J stage) and samples at the F, A, and P stages, indicating that the metabolites considerably changed during the whole black mulberry winemaking process.

OPLS-DA was used to analyze the metabolite variation and maximize the difference between groups during the fermentation process. It is more reliable, as OPLS is an orthogonal rectification to PLS-DA that filters out unrelated orthogonal signals. In total, four groups were distinguished in the model: J vs. F, F vs. A, A vs. P, and J vs. P (Figure 2C–F). Metabolite changes were greater during fermentation (F) than ripening (A) and black mulberry wine (P). The score plot showed excellent model parameters (J vs. F:R^2^Y = 1, Q^2^ = 0.994; F vs. A:R^2^Y = 1, Q^2^ = 0.99, A vs. P:R^2^Y = 0.999, Q^2^ = 0.975, J vs. P:R^2^Y = 1, Q^2^ = 0.995), respectively. Additionally, cross-validation and replacement tests showed that the OPLS-DA model was not over-fitted (Appendix A). These results demonstrate that the OPLS-DA model can be effective in exploring metabolic differences at different stages of the mulberry wine making process.

### 3.3. Screening of Differential Metabolites

#### 3.3.1. Determination of Differential Markers

The variable importance in the projection (VIP) of the first principal component of the OPLS-DA model with the *p*-value of the student’s *t*-test was used to identify differential metabolites in this study. Metabolites of four groups (J vs. F, F vs. A, A vs. P, and J vs. P) were screened in the positive and negative ion modes according to VIP > 1 and *p* < 0.05. A total of 1597 (669 up-regulated; 928 down-regulated) differential metabolites (Appendix A) were identified in the four groups during the ESI+ and ESI− patterns (J vs. F vs. A vs. P). Meanwhile, hierarchical cluster analysis (HCA) was used to compare the groups, calculate the quantitative values of the differential metabolites, and reveal the diversity of metabolite profiles at different fermentation stages (Appendix A).

Furthermore, to gain a deeper understanding of the variations in primary metabolites during mulberry wine fermentation, the principal differentially expressed metabolites were sorted based on their VIP values. The organic heterocyclic compounds, amino acids, terpenoids, phenylpropanoids, alkaloids, aromatic compounds, carbohydrates, glycerol phospholipids and triglycerides, flavonoids, lignin, isoflavones, organic acids, fatty acids, steroids, etc., were identified during mulberry wine fermentation (Figure 3). Notably, the metabolic properties of the primary metabolites exhibited a comparative variance during diverse stages of the fermentation process. The differential metabolites primarily manifested in their relative abundance and species, which corresponded to the hierarchical cluster analysis (HCA) findings (Appendix A). Specifically, they played a crucial role in the quality parameters of mulberry wine, such as flavonoids and amino acids interacting with volatiles and polyphenols determining the sensory qualities of grape wine [22].

In our study, among the categories with higher content, polyphenols are noted. The essential role of these in the sensory and nutritional characteristics of wine has been demonstrated by previous research [23,24,25]. In this work, the most diverse metabolites with differential production identified in the database belong to polyphenols, particularly flavonoids (78 in total), and they may be considered the main actors in the quality features of mulberry wine. Therefore, to enhance our understanding of the fluctuations in flavonoid profiles during the mulberry fermentation process, we selected compounds linked with flavonoids that showcase secondary metabolite signals for further examination.

#### 3.3.2. Relationships between Major Metabolites and Physicochemical Properties

Polyphenols and amino acid metabolite composition were correlated with the total sugar content and alcohol concentration (r = 0.76; r = 0.86 and r = 0.58; r = 0.68, respectively) during the mulberry fermentation process (Figure 4; Appendix A). Furthermore, the composition of aromatic compounds and organic acids (r = 0.84, r = 0.69, respectively) metabolites were strongly correlated with the total sugar content during the whole winemaking process. However, there was no significant correlation between pH, TA, and TSS factors and the composition of all major metabolites (Mantel’s *p* > 0.1) (Figure 4; Appendix A). The present study results suggested that the composition of major metabolites (polyphenols, amino acids, aromaticity, fatty acids, organic acids, and carbohydrates) was related to some physiochemical characteristics (TS, alcohol) (Figure 4). This is because of the interaction between ethanol, total sugar, and oxygen can generate reactive oxygen species, causing oxidative stress to microorganisms such as yeast and affecting their utilization of precursor substances, leading to changes in metabolic activity [26]. Additionally, stable complexes that affect the solubility of amino acids, polyphenols, aromatic compounds, and organic acids are produced as a result of this interaction, which in turn leads to changes in their concentrations [22,27]. Notably, the total sugar and alcohol concentrations were closely related to the composition of a new dimension and a more comprehensive investigation of major metabolites in the mulberry fermentation process.

### 3.4. Flavonoid Metabolites and Flavonoid Metabolic Pathway Analysis

The total amount of flavonoids decreased by 7.7-fold. Six flavonoid metabolites in Table 2 were identified during the fermentation process of mulberry with differential metabolite markers based on VIP score > 1. The flavonoid changes in the fermentation stage were consistent with the black mulberry wine product. Especially, three flavonoid metabolites, such as luteolin, luteolin-7-O-glucoside, and (−)-epiafzelechin, were up-regulated while eriodictyol was down-regulated in fermented period. Kaempferol identified in both fermentation, ripening, and the black mulberry wine product period was also upregulated, which was consistent with quercetin found in the ripening period. Luteolin and luteolin-7-O-glucoside were mostly found in the apple peel and carrot, which exhibited protective effects against oral cancer [28], and (−)-epiafzelechin was found in many herbal medicines, such as tea tree and Senna obtusifolia [29], and luteolin, and luteolin-7-O-glucoside were significantly up-regulated during the winemaking process, increasing by 2.07 and 7.85-fold during the fermentation period and up to 5.59 and 13.57 over the whole fermentation process, respectively. (−)-Epiafzelechin was up-regulated by 1.73-fold during fermentation and up to 3.21 throughout the fermentation process. Kaempferol is the primary flavonoid in citrus fruits and has anti-inflammatory and antioxidant properties [30]. In this study, kaempferol increased by 8.58-fold in the fermentation period and decreased by 2.04-fold during the whole fermentation period.

The changes in eight flavonoid metabolites’ concentrations are depicted in Figure 5. Among them, (kaempferol 3,4′,5,7-tetrahydroxyflavone, luteolin 7-O-glucoside, quercetin 3,3′,4,5,7-pentahydroxyflavone, and (−)-epiafzelechin 3,5,7,4′-tetrahydroxyflavan) were significantly reduced during the process (*p* < 0.01). Kaempferol, luteolin-7-O-glucoside, and quercetin are all structure-based 2-phenyl chromogranin, which reacted with the hydroxyl groups during the alcoholic fermentation period, thereby decreasing the metabolites’ concentration [31]. Additionally, the structural stability of quercetin was influenced by pH, temperature, metal ions, and other compounds [32,33]. Given the intricacies involved in the biochemical processes of winemaking, alterations in the environmental conditions and microbial communities of the fermentation system can significantly impact the resulting flavonoid profiles. Such changes can manifest in diverse chemical compositions and thus affect the sensory attributes of the final product. The other three flavonoid metabolites (eriodictyol (S)-3′,4′,5,7-tetrahydroxyflavanone (*p* < 0.01), epicatechin (−)-3,3′,4′,5,7-pentahydroxyflavan (*p* < 0.5), rutin (3,3′,4′,5,7-pentahydroxyflavone-3-rhamnoglucoside) (*p* < 0.01) increased in the fermentation process and then decreased in the ripening and production processes (Figure 5). Epicatechin was identified as the most abundant flavonoid metabolite, which increased due to microbial fermentation, and decreased due to the condensation or oxidation reactions during the ripening process [34,35]. In a previous study, eriodictyol and rutin exerted multiple biological activities, including antivirus, antioxidation, and anti-inflammation effects [36,37]. In the fermentation period, microbial interaction can influence the extraction of flavonoids from mulberry fruits [8]. However, the profiles of eriodictyol and rutin significantly altered during ripening or final mulberry wine processing, which were indicative of additional condensation, esterification, and oxidation reactions due to the addition of sulfur dioxide [38,39]. In contrast, an opposite trend was observed for luteolin (3′,4′,5,7-tetrahydroxyflavone) compared with the above three metabolites. Overall, the flavonoid metabolites identified during various fermentation stages exhibited a diverse range of content, as shown in Figure 5. These metabolites also displayed varying VIP values and FC, as reported in Table 2. Furthermore, the intricate biochemical process underlying the production of these metabolites is likely responsible for the observed variability. This finding highlights the need for a deeper understanding of the underlying biochemical mechanisms that influence flavonoid metabolite formation during fermentation.

Enrichment analysis of the metabolic pathway, an effective method, clarified metabolites based on the KEGG database in mulberry wine fermentation. A total of 96 metabolic pathways (involving 1597 different metabolites) were identified throughout the mulberry fermentation process. Enrichment analysis was used to analyze the top 15 differential metabolite pathways (Appendix A). After careful analysis, it was discovered that two metabolic pathways related to flavonoids played a significant role throughout the mulberry winemaking process. Specifically, these pathways encompassed flavonoid biosynthesis (https://www.genome.jp/kegg-bin/show_pathway?ko00941/ko:C12128%09red/ko:C01514%09green/ko:C05631%09green/ko:C05903%09red/ko:C00389%09red/ko:C09727%09green; accessed on 8 May 2023), as well as flavone and flavonol biosynthesis (https://www.genome.jp/kegg-bin/show_pathway?ko00944/ko:C01514%09yellow/ko:C03951%09yellow/ko:C05903%09blue/ko:C00389%09blue/ko:C05625%09blue; accessed on 8 May 2023).

Flavonoid metabolites were mainly derived from phenylpropanoid biosynthesis. Overall, phenylalanine produced p-coumaroyl-CoA by L-phenylalanine biosynthesis, in which p-coumaroyl-CoA was metabolized into several flavonoids (kaempferol, quercetin, (−)-epiafzelechin, epicatechin, luteolin, and eriodictyol) through the flavonoid biosynthesis pathway [40]. On the other hand, p-coumaroyl-CoA produced luteolin, luteolin-7-O-glucuronic acid, quercetin, and rutin through the biosynthesis of flavone and flavonols pathway (Figure 5). Notably, quercetin and luteolin were involved in both flavonoid biosynthesis and flavone and flavonols biosynthesis.

Generally, multiple metabolic pathways in the winemaking process involve various enzymes. For instance, the enzymes (chalcone synthase, phenylalanine ammonia-lyase, and flavanone-3-hydroxylase) involved in flavonoid biosynthesis are related to the biosynthesis and derivatives of kaempferol [41]. In this study, chalcone isomerase, naringenin 3-dioxygenase, flavonol synthase, and flavonoids 3′,5′-hydroxylase catalyzed caffeoyl-CoA to produce quercetin. Then, the flavonol 3-O-glucosyltransferase and flavonol 3-o-glucoside L-rhamnosyl transferase catalyzed quercetin to the pyrannosyl and pyranosyl groups to form rutin. Meanwhile, caffeoyl-CoA was catalyzed to produce of luteolin through chalcone synthase, flavone synthase I, and flavone synthase II in flavonoid biosynthesis [42].

## 4. Conclusions

In this study, higher-level differential metabolites were discovered, including organic heterocyclic compounds, amino acids, phenylpropanoid compounds, aromatic compounds, and carbohydrates (top 5). These findings highlight significant differences in the abundance and concentration of metabolites during different stages of winemaking. Additionally, the relationships between major metabolites and physicochemical factors indicate that total sugars and alcohol are associated with the formation of polyphenolic compounds during the fermentation process. The metabolism of polyphenolic compounds, especially flavonoids (78 types), such as luteolin, luteolin-7-O-glucoside, (−)-epiafzelechin, eriodictyol, kaempferol, and quercetin, serve as differential metabolite markers that exhibit significant changes during the fermentation and ripening of mulberry wine. A total of 96 metabolic pathways were identified, among which flavonoid, flavone, and flavonol biosynthesis were the major transformation pathways of flavonoid compounds. Overall, our research provides new insights into the dynamic changes in flavonoid profiles during black mulberry winemaking. These findings could be useful for optimizing the winemaking process and developing new black mulberry-based products with enhanced health benefits.

## Figures and Tables

**Figure 1 foods-12-02221-f001:**
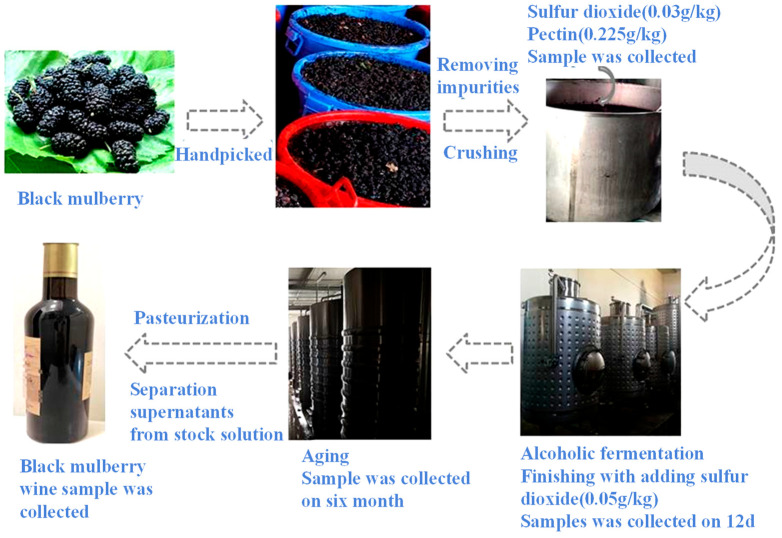
Schematic diagram of mulberry wine fermentation process.

**Figure 2 foods-12-02221-f002:**
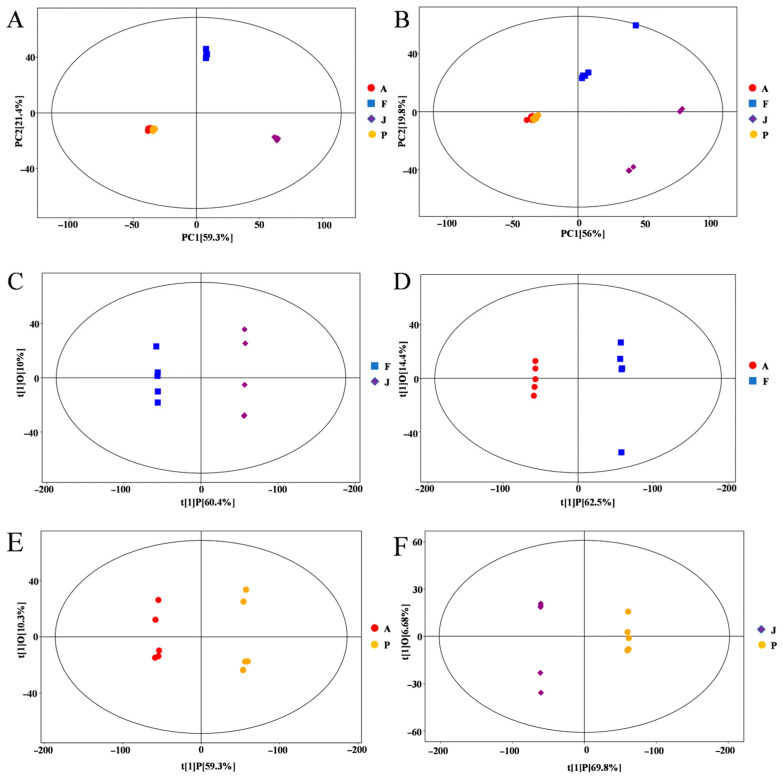
Principal component analysis (PCA) score plots of samples and quality control (QC) samples at different stages of the mulberry fermentation process in ESI+ (**A**) and ESI− (**B**) patterns. OPLS-DA in positive ion mode yields scatterplot: (**C**) J vs. F; (**D**) F vs. A; (**E**) A vs. P; and (**F**) J vs. P.

**Figure 3 foods-12-02221-f003:**
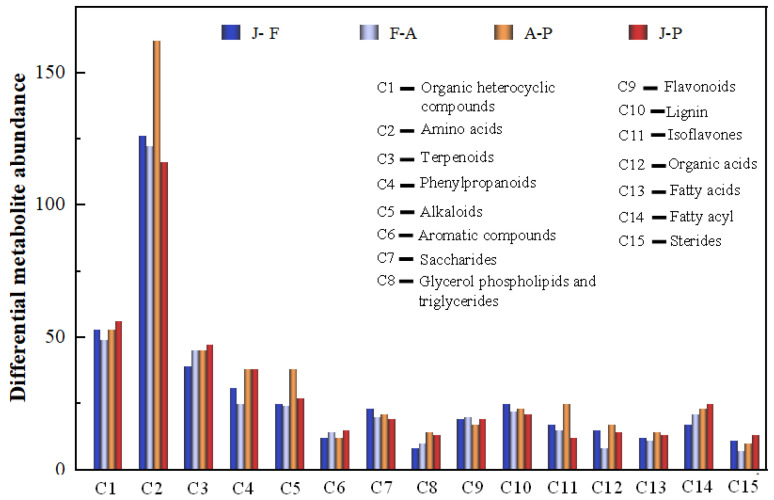
Different metabolites in the fermentation of mulberry wine. Analysis of main metabolite (top 15) in different stages of mulberry wine fermentation (blue: J-F (Fermentation); light steel-blue: F-A (ripening); orange: A-P (production); red: J-P (mulberry juice to wine).

**Figure 4 foods-12-02221-f004:**
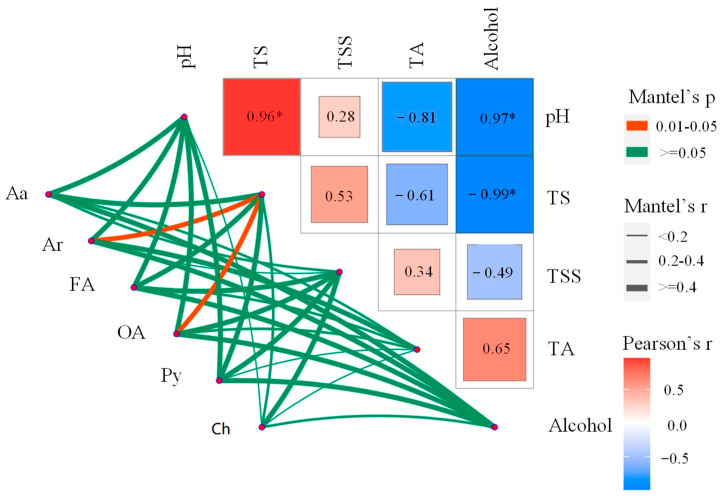
Analysis of the association between major differential metabolites and environmental factors using the mantel test. A color gradient is used to represent the pairwise comparison of physicochemical factors in Spearman’s correlation coefficient. Mantel’s test: amino acids (Aa), aromaticity (Ar), fatty acids (Fa), organic acids (Oa), polyphenols (Py), and carbohydrates (Ch) were correlated with environmental factors during the fermentation of mulberry wine. Edge width corresponds to the r statistic of Mantel for the corresponding distance correlation, and edge color represents statistical significance based on 9999 permutations. *: Significance maker.

**Figure 5 foods-12-02221-f005:**
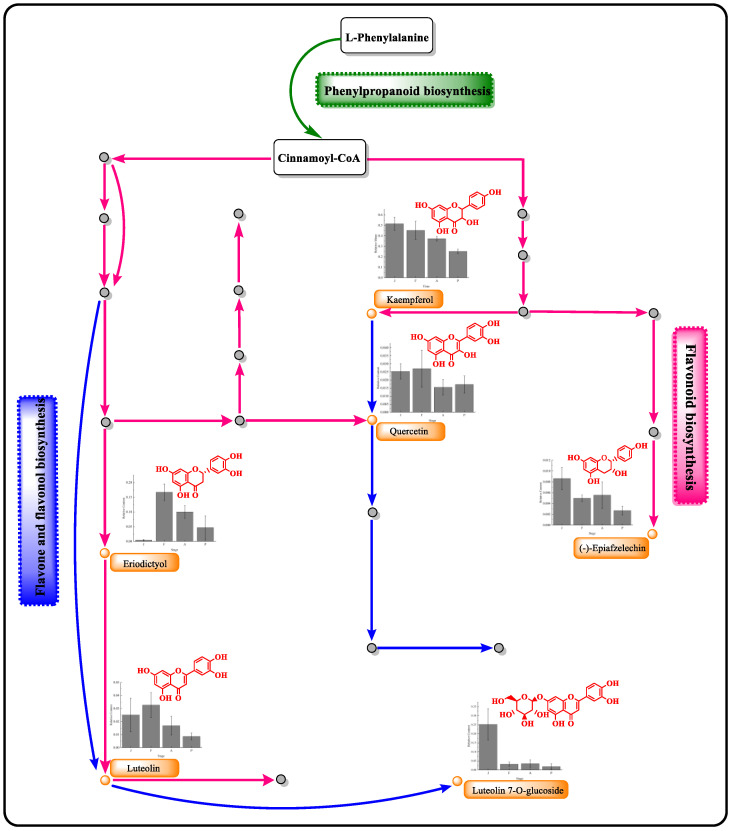
Variation in six flavonoid contents and related metabolic pathways during the fermentation of mulberry wine. (Blue: Phenylpropanoid biosynthesis pathway; green: flavone and flavonol biosynthesis pathway; black: flavonoid biosynthesis pathway).

**Table 1 foods-12-02221-t001:** Physiochemical indices of black mulberry winemaking process.

Physicochemical Indices	Black Mulberry Winemaking Process			
	J	F	A	P
pH	5.48 ± 0.006 _a_	3.92 ± 0.006 _c_	3.99 ± 0.006 _d_	3.95 ± 0.010 _b_
TA (g/L tartaric acid)	2.793 ± 0.326 _c_	12.757 ± 2.295 _a_	6.597 ± 0.078 _b_	8.443 ± 0.078 _b_
TS (g/L)	139.152 ± 26.492 _a_	58.889 ± 1.111 _b_	21.783 ± 0.729 _c_	39.974 ± 2.884 _cb_
TSS (%)	16.9 ± 0.100 _b_	19.97 ± 0.153 _a_	11.33 ± 0.578 _d_	12.43 ± 0.252 _c_
Alcohol (*v*/*v*, %)	0 _c_	8.0 ± 0.1 _b_	10.0 _a_	10.0 ± 0.1 _a_

Data are presented as means ± standard deviation (*n* = 3). TS: total sugar; TA: total acid; TSS: total soluble solid; different lowercase letters (a, b, c, d) indicate significant differences between the samples (*p* < 0.05, one-way ANOVA).

**Table 2 foods-12-02221-t002:** Flavonoid metabolite markers with the strongest distinguish ability at different fermentation stages.

Fermentation Stage	Marker Compound	RT (min)	*m*/*z*	Mass Error	Ion Model	VIP	*p*-Value	FC	log_2_ (FC)
J-F	Luteolin	5.0729	287.0547	0.9783	pos	1.1786	0.0030	2.0701	1.0497
	Luteolin-7-O-glucoside	8.9374	449.1077	0.9403	pos	1.2407	0.0044	7.8548	2.9736
	(−)-Epiafzelechin	2.4847	275.0911	0.8052	pos	1.0672	0.0145	1.7339	0.7940
	Eriodictyol	6.6508	287.0538	0.7186	neg	1.4765	0.2353	0.0357	−4.8081
F-A	Luteolin	5.0729	287.0547	0.9783	pos	1.1847	0.0001	0.5665	−0.8198
	Kaempferol	0.7897	285.0407	0.9975	neg	1.1712	0.2597	8.5817	3.1013
	Quercetin	1.0179	301.0354	0.9579	neg	1.0688	0.2721	7.5019	2.9072
A-P	Kaempferol	2.9898	287.0545	0.9975	pos	1.2480	0.0000	1.4801	0.5657
	Luteolin	5.0624	287.0547	0.9783	pos	1.2650	0.0000	4.7709	2.2543
J-P	Kaempferol	2.9898	287.0545	0.9975	pos	1.1606	0.0000	2.0471	1.0336
	Luteolin	5.0729	287.0547	0.9783	pos	1.1650	0.0005	5.5954	2.4842
	Luteolin-7-O-glucoside	8.9374	449.1077	0.9403	pos	1.1403	0.0033	13.5785	3.7632
	(−)-Epiafzelechin	2.4847	275.0911	0.8052	pos	1.1044	0.0003	3.2103	1.6827

VIP: variable importance in projection; FC: fold change; mass error: the scoring of the secondary match and takes the value [0, 1], closer to 1 means higher accuracy.

## Data Availability

All data needed to evaluate the conclusions in the paper are present in the paper and/or the Appendix A.

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
