# Peer review of "Metabolomics-Based Analyses of Dynamic Changes in Flavonoid Profiles in the Black Mulberry Winemaking Process"

_foods, 2023, doi:10.3390/foods12112221_

Round 1
Reviewer 1 Report
The manuscript presents interesting data about metabolomic variation in black mulberry wine at different procedure stages, with a focus on polyphenol fraction. However, the manuscript is not clearly written, not easy to read, and lacking in content, however, some points should be improved as suggested below:
1. Generally, the English level of the paper is poor. Please improve it.
2. I suggest substituting “aging” with “ripening”.
3. Please, add section 2.1 “Chemical and Reagent” in Materials and Methods.
4. I recommend to move the section “Collection of Samples” before the “Mulberry winemaking”.
5. Reformulate lines 100-101 to better explain the abbreviations used for samples.
6. It is well known that polyphenols are extracted with polar solvents (water, methanol, ethanol). Could you explain the choice to use ACN? Please, report the reference for the extraction method.
7. I do not understand the organization of the sections. For example, metabolite extraction could be combined with a collection of samples; UHPLC-QE-MS/MS analyses with data processing.
8. In the section of UHPLC-QE-MS/MS analyses, should report the gradient of the separation, the analysis time, flow rate, column temperature and mass range.
9. In general, conventional polyphenol separations are carried out with a C18 column and as mobile phases water and acetonitrile acidified with formic or acetic acid. Explain why you chose an amide column and mobile phases with ammonium acetate and ammonia hydroxide in water that are not really green. Some references for polyphenols: Molecules 2021, 26, 2710. https://doi.org/10.3390/molecules26092710, Molecules 2023, 28(4), 1865; https://doi.org/10.3390/molecules28041865.
10. Please (v/v), versus and m/z in italic.
11. Table 1: Please, report the meaning of the abbreviations (TA, TS, TSS, c,b,a). It is difficult to read the results.
12. Please, show the chromatogram for the most complex sample or one with a comparison of the samples analysed.
13. In figure 2 why you didn’t plot F vs. P also?
14. Please, report the mass error in table 2 and in the table of supplementary material.
15. Line 231: add some comments about the huge number (1597 real??) of different metabolites that you have found.
16. Figure 3 and lines 239-241: I’m very skeptical that through analysis and extraction (ACN:MeOH) you could find all these chemical classes. Please explain in more detail in the text this.
17. Lines 249-251: should be added other references about polyphenols role. Molecules 2021, 26, 2710. https://doi.org/10.3390/molecules26092710, Molecules 2023, 28(4), 1865; https://doi.org/10.3390/molecules28041865.
18. Add the measurement unit used for quantification in section 3.4 (Line 290), is the fold change the best way to explain these results? What are 991.37 and 128.58???
19. Table 2: What do VIP and FC mean? Please explain in the legend.
20. How long is the HPLC analysis? Looks really long from the retention time in Table 2 (9 hours?????? it is impossible).
Please, check the links in lines 349-354, they do not work.
Generally, the English level of the paper is poor. Please improve it.
Reviewer 2 Report
In the current manuscript, the metabolic profiles and particularly the flavonoid profiles of Mulberry throughout the process of vinification were studied using the UHPLC-QE-MS/MS methodology coupled with multivariate statistical analyses. The paper has a good scientific writing structure and innovative topic. Appropriate methods have been used to compile this article.
In my opinion, it has a good quality in the Foods.
My comments are as follows:
L26: Do not use the words used in the title in the keywords.
L42: write the Lactic acid bacteria in italic form.
-You have used black mulberry in this research, but you have not mentioned its scientific name anywhere in the text.
L74-75: Write the time (season) of harvesting the fruits.
L75: How did you crush the fruits? Write the specifications of the device used?
L78: write Saccharomyces cerevisiae in italic.
L87: write the time of pasteurization.
L102: write "respectively" at the end of sentence.
L107: write the unit of total acid. For example, as g/L citric or acetic acid.
L109: remove "°" from the top of Brix°.
L110: Brix meter or Refractometer? Write the specific of the device.
L116-117: rewrite these sentences and mention more details.
L189: Table 1. Use the correct letters to show the significance among the treatments. The letters in front of pH are incorrect (recheck and rewrite). Also, change 12.43±0.252d to 12.43±0.252c and 11.33±0.578c to 11.33±0.578d.
Finally, do you recommend this method for fermenting other fruits?
Round 2
Reviewer 1 Report
The paper can be accepted in present form.